# The Survey Measure of Psychological Safety and Its Association with Mental Health and Job Performance: A Validation Study and Cross-Sectional Analysis

**DOI:** 10.3390/ijerph19169879

**Published:** 2022-08-11

**Authors:** Natsu Sasaki, Akiomi Inoue, Hiroki Asaoka, Yuki Sekiya, Daisuke Nishi, Akizumi Tsutsumi, Kotaro Imamura

**Affiliations:** 1Department of Mental Health, Graduate School of Medicine, The University of Tokyo, Tokyo 113-0033, Japan; 2Department of Public Mental Health Research, National Institute of Mental Health, National Center of Neurology and Psychiatry, Kodaira 187-8553, Japan; 3Institutional Research Center, University of Occupational and Environmental Health, Kitakyushu 807-8555, Japan; 4Department of Psychiatric Nursing, Graduate School of Medicine, The University of Tokyo, Tokyo 113-0033, Japan; 5Department of Public Health, School of Medicine, Kitasato University, Sagamihara 252-0374, Japan

**Keywords:** occupational health, leadership, mental health, workplace climate, worksite

## Abstract

Objectives: This study validated the Japanese version of O’Donovan et al.’s (2020) composite measure of the psychological safety scale and examined the associations of psychological safety with mental health and job-related outcomes. Methods: Online surveys were administered twice to Japanese employees in teams of more than three members. Internal consistency and test–retest reliability were tested using Cronbach’s α and intra-class correlation coefficient (ICC), respectively. Structural validity was examined using confirmatory factor analysis (CFA) and exploratory factor analysis (EFA). Convergent validity was tested using Pearson’s correlation coefficients. Multiple linear regression analyses were conducted to examine the relationship between psychological safety and psychological distress, work engagement, job performance, and job satisfaction. Results: Two hundred healthcare workers and 200 non-healthcare workers were analyzed. Internal consistency, test–retest reliability, and convergent validity were acceptable. CFA demonstrated poor fit, and EFA yielded a two-factor structure, with team leader as one factor and peers and team forming the second factor. The total score showed significant and expected associations with all outcomes in the adjusted model for all workers. Conclusions: The Japanese version of the measure of the psychological safety scale presented good reliability and validity. Psychological safety is important for employees’ mental health and performance.

## 1. Introduction

Psychosocial factors at work are well-known determinants of workers’ health and well-being. Psychological safety (PS) at work has received much attention as an important psychosocial factor in workers’ positive mental health and other work-related outcomes, such as work engagement, satisfaction, communication, and performance [1,2]. PS describes workers’ perceptions of the consequences of taking interpersonal risks in a particular context, such as a workplace [3,4]. In 1999, Edmondson defined PS as a shared belief that the team is safe for interpersonal risk-taking (i.e., doing learning behavior that may place workers at risk, including seeking feedback, sharing information, asking for help, talking about errors, and experimenting) [3].

Previous review articles have reported three streams of research on PS (i.e., individual-, team-, and organizational-level), with team-level analysis the largest and most active [1,4]. A meta-analysis has reported that individual- and team-level PS is significantly related to work engagement, task performance, information sharing, creativity, learning behavior, and job satisfaction [2]. Recent studies have investigated the mediating role of PS in the association of leadership with job performance and mental health [5,6,7,8]. Papers published in the 2020s have focused more on the mediating effect of PS in the relationship between, for example, supervisor–subordinate communication and employees’ commitment [9], employees’ positive affect and motivations [10], and cognitive stress and turnover intentions [11]. Thus, accumulating evidence suggests that PS (especially individual- and team-level PS) is important for workers’ health and well-being. In Japan, the concept of PS is becoming increasingly popular, along with growing interest in health and productivity management (H&PM), and it is expected that improving PS will enhance employees’ mental health and performance. Nevertheless, epidemiological research on PS has not progressed sufficiently due to the lack of a multidimensional PS scale.

Many studies have used self-reported questionnaires adapted from Edmondson’s team-level measure to quantitatively assess PS at work [1]. Although several scales with fewer than 10 items can measure PS in non-healthcare workers (non-HCWs) [3,12,13,14], including the Japanese version of the PS scale [12] developed by Liang et al. [15], multidimensional measurement of the individual and team levels of PS is unavailable. O’Donovan et al. (2020) presented a 19-item composite measure of PS (i.e., observation and survey component) containing three subsections (i.e., team leader, peers, and team) for use by healthcare teams, which they co-developed with healthcare professionals based on six measures and the PS literature [16]. The 19 items were identified as the corresponding comprehensive behaviors relevant to PS [16]. The three sections (i.e., team leader, peers, and team) were based on the real voices of professionals in the clinical settings, which revealed that the difficulty of taking actions related to PS was different for superiors or peers. The three sections that assess the individual and team levels of PS could provide detailed information about PS. A systematic review suggested that scales with a few items could not fully capture the state of PS at work; therefore, holistic, objective measuring instruments are needed [17]. A multidimensional and scalable measure could thus be used to investigate the association of these three components with employees’ mental health and performance and to develop an effective intervention plan, among the variety of the workers, including HCW and non-HCW groups.

The associations of the individual and team levels of PS with mental health and work-related outcomes have not been investigated yet in HCW and non-HCW groups. A previous systematic review presented possible pathways from job resources (e.g., supportive leadership behavior) to positive and negative work outcomes (e.g., stress, conflict, and performance) through PS in the integrative theoretical framework of PS [1]. Some previous studies have suggested that PS reduced the risk of poor mental health outcomes, such as burnout, stress, and diminished well-being, by increasing social support for HCW and non-HCW [18,19]. However, the effect of PS on mental health has not been empirically examined. In addition, the effect may be different in HCWs and non-HCWs because the clinical settings offer different working conditions. Further study is needed to investigate the association of PS at work with mental health in both groups of workers using well-developed measures of PS.

The objectives of this study were: (i) to develop the Japanese version of the survey measure of PS introduced by O’Donovan et al. (2020) [16] and examine internal consistency, test–retest reliability, structural validity, and convergent validity of the scale in HCWs and non-HCWs; and (ii) to examine the associations of PS with psychological distress, work engagement, job performance, and job satisfaction.

## 2. Method

### 2.1. Scale Information and Participants

Although the measures developed by O’Donovan et al. (2020) were tailored to healthcare settings, the survey measure of PS could also be useful for measuring PS in non-HCWs. We obtained permission from O’Donovan, the developer of the original scale, to translate the measures into Japanese and validate them in HCWs and non-HCWs. The scale has 19 items divided into three sections (i.e., team leader, peers, and team), as introduced earlier. The Japanese version of the survey measure of PS was developed according to the procedure specified in the International Society of Pharmacoeconomics and Outcomes Research (ISPOR) task force guidelines [20]. The forward translation was conducted independently by two external translators proficient in Japanese and English. We then performed reconciliation, back-translation, back-translational review, harmonization, and cognitive debriefing. NS and YS conducted reconciliation, and KI chose the appropriate expression of the items. A native English translator back-translated the scale unaware of the original scale. The original developer confirmed and accepted the back-translated measures. Cognitive debriefing sessions were conducted with three Japanese nurses, including HA Their feedback about difficult wording was used for further modifications. The results from these stages were combined to develop the final measure. The full Japanese version of the survey measure of PS is presented in Appendix A. The final scale contained 19 items, with nine items for the team leader, seven items for peers, and three items for the team as a whole, measured on a seven-point Likert scale. The scale score was calculated by averaging the items. Higher scores indicated greater PS.

Online surveys were administered twice to Japanese employees who had not been appointed as leaders of their team at baseline (January 2022) and at a two-week follow-up (February 2022). The Research Ethics Committee of the Graduate School of Medicine/Faculty of Medicine, The University of Tokyo, approved the study, No. 2019361NI-(3). The study was reported according to the Consensus-based Standards for the Selection of Health Measurement Instruments (COSMIN) guideline, which is used to improve the quality of efforts to develop health-related self-report measurement instruments [21].

Participants living in Japan were invited from the registered panel of an Internet research company (Rakuten Insight Inc., Tokyo, Japan). Equal numbers of HCW and non-HCW were recruited. Participants’ inclusion criteria were as below:(i)full-time employees 20–65 years old;(ii)working for a company with more than five employees;(iii)joined a team with more than three members;(iv)not a president or manager;(v)not a team leader.

All participants at baseline were invited to participate in a two-week follow-up. The follow-up survey was closed after 100 answers were collected.

### 2.2. Measurements

To test the convergent validity, the psychological safety scale for workers developed by Liang et al., social support at work, servant leadership, organization-based self-esteem, and organizational justice were measured.

Psychological safety was measured with the PS scale developed by Liang et al. (2012) that reflects Kahn’s [22] focus on the workers’ speaking out [15]. The Japanese version of the scale was translated by Ochiai et al. [12]. It contained five items measured on a five-point Likert scale. The items asked workers to rate the extent to which they feel free to express their thoughts and feelings. The scale score was calculated by averaging the items. Higher scores indicated greater PS. Cronbach’s alpha was 0.71 in this sample.

Social support at work was measured using the Brief Job Stress Questionnaire (BJSQ) [23] containing items assessed on a four-point Likert scale. Social support at work comprises two subscales: supervisor support (three items) and co-workers’ support (three items). A higher score indicated higher social support at work. In this sample, Cronbach’s alphas were 0.89 for supervisor support and 0.88 for co-workers’ support.

Servant leadership was measured with the Japanese short version of the Servant Leadership Survey (SLS-J) [24] evaluating the employees’ supervisors. This scale includes six items measuring empowerment (leader side), three items measuring humility (servant side), three items measuring standing back (servant side), three items measuring stewardship (leader side), and three items measuring authenticity (servant side) on a six-point Likert scale. The score for each dimension of the SLS-J-short was calculated by averaging the item scores. A higher score indicated stronger servant leadership. Cronbach’s alpha was 0.95 for empowerment, 0.91 for humility, 0.84 for standing back, 0.83 for stewardship, and 0.81 for authenticity.

Organization-based self-esteem was measured using the Japanese version of the Organization-based Self-Esteem Scale [25]. This scale has eight items measured on a five-point Likert scale. The scale score was calculated by averaging the items. A higher score indicated higher organization-based self-esteem. Cronbach’s alpha was 0.94.

Organizational justice was measured with the Japanese version of the Organizational Justice Questionnaire (OJQ) [26]. The OJQ consists of two subscales: procedural justice and interactional justice. Seven items assess procedural justice, and six items assess interactional justice on a five-point Likert scale. Each factor score was calculated by averaging the items. A higher score indicated a greater degree of organizational justice. Cronbach’s alpha was 0.93 for procedural justice and 0.95 for interactional justice.

To examine the associations of the PS scale with mental health and job-related outcomes, psychological distress, work engagement, job performance, and job satisfaction were measured.

Psychological distress was measured with the Japanese version of the K6 scale [27,28]. This scale has six items (felt nervous, hopeless, restless or fidgety, worthless, depressed, and that everything was an effort in the past four weeks) rated on a five-point Likert scale. The total score was calculated by summing all items. The higher score indicated greater distress. Cronbach’s alpha was 0.93.

Work engagement was measured using the Japanese version of the Utrecht Work Engagement Scale (UWES-9) [29]. This scale has nine items rated on a seven-point Likert scale. The scale score was calculated by averaging the items. The higher score indicated greater work engagement. Cronbach’s alpha was 0.96.

Work performance was evaluated using one item of the Japanese version of the WHO Health and Work Performance Questionnaire (HPQ) [30]. Participants were asked to rate their work performance over the past four weeks. Items were scored on an 11-point scale ranging from 0 (worst) to 10 (best). A high score indicated good work performance.

Job satisfaction was measured by one item from the Brief Job Stress Questionnaire (BJSQ) [23] on a four-point Likert scale. A higher score indicated more job satisfaction.

Demographic variables were gender, age, education attainment, working from home, marital status, company size, occupation (e.g., professions, service workers), and job category (e.g., doctor, nurse) at baseline.

### 2.3. Statistical Analysis

In this study, the HCWs and non-HCWs were analyzed separately. First, the distribution of demographic characteristics as well as means and standard deviations (SDs) for the total scores of the PS scale and its three subscales at baseline and follow-up were calculated. Then, to assess internal consistency and test–retest reliability of the PS scale, Cronbach’s α and intra-class correlation coefficient (ICC) for each of the subscales were calculated, following the COSMIN guidelines [21]. To assess structural validity, a confirmatory factor analysis (CFA) with three factors (i.e., team leader, peers, and team) was conducted to test the goodness of fit of the existing structure of PS. Model fit was assessed using a combination of fit indices including the chi-square (χ^2^), the comparative fit index (CFI), the Tucker-Lewis index (TLI), the root mean square error of approximation (RMSEA), the standardized root mean square residual (SRMR), the goodness of fit index (GFI), the Akaike’s information criterion (AIC), and the adjusted goodness of fit index (AGFI). If the CFA showed a poor fit, an exploratory factor analysis (EFA), which hypothesized no factor structure with the Promax rotation method, using a robust maximum likelihood estimation, was conducted. To test the hypotheses (expected relationships with other outcomes), convergent validity was examined using Pearson’s correlation coefficients (r) which were calculated between each score of the PS scale and PS scale for workers developed by Liang et al., social support at work, servant leadership, organization-based self-esteem, and organizational justice, which was considered to have moderate to high positive correlations with PS scale (r > 0.40) [12].

Since both independent and dependent variables were continuous, we conducted multiple linear regression (MLR) analyses to examine the relationship between the PS scale and outcomes (i.e., psychological distress, work engagement, job performance, and job satisfaction). After standardizing these variables, we first examined crude associations. Second, we examined adjusted associations considering the covariates for gender, age, educational attainment, working from home, marital status, company size, occupation, and job category simultaneously. Previous studies related to PS have frequently used MLR analysis [31,32], and this study followed traditional formulas [33,34] to estimate the relationship between theoretically and practically related variables. As literature suggested [1,2], PS can influence outcomes investigated in this study theoretically and conceptually. In addition to the full scale, we examined the relation of three subscales, putting each scale in the model individually (Model 1) and simultaneously (Model 2).

Statistical significance was defined as *p* < 0.05. IBM SPSS Statistics^®^ version 28 (IBM, Armonk, NY, USA) and IBM SPSS Amos^®^ version 28 were used for the analyses.

## 3. Results

The demographic characteristics of 400 participants (200 HCW and 200 non-HCW) are presented in Table 1. Among HCWs, 60% of participants were women, 58% were married, and 90% were employed in the medical industry. The mean age was 40.1 (SD = 9.6). HCWs included physicians (14%), nurses/midwives/public health nurses (48%), and others (39%). The number of team members was 20 or more (45%), 11–19 (23%), and 6–10 (21%). Among non-HCWs, 69% of the participants were men, 57% were married, and 25% were employed in the manufacturing industry. The mean age was 43.4 (SD = 10.7). The number of team members was 6–10 (44%), 3–5 (29%), and 11–19 (15%).

Internal consistency and test–retest reliability values of the PS scale are presented in Table 2. For HCWs, the Cronbach’s alpha of each section ranged from 0.91 to 0.95, ICC ranged from 0.75 to 0.89, the mean total score was 4.96, and Cronbach’s alpha was 0.96. For non-HCWs, Cronbach’s alpha ranged from 0.93 to 0.96, ICC ranged from 0.84 to 0.92, the mean total score was 4.63, and Cronbach’s alpha was 0.92.

The results of confirmatory factor analyses were χ^2^ (149) = 540.001, CFI = 0.899, TLI = 0.884, RMSEA = 0.115, SRMR = 0.0444, GFI = 0.764, AIC = 622.001, and AGFI = 0.699 for HCWs. For non-HCWs, the values were χ^2^ (149) = 584.778, CFI = 0.903, TLI = 0.888, RMSEA = 0.121, SRMR = 0.0472, GFI = 0.733, AIC = 666.778, and AGFI = 0.659. Factor loadings for each item of PS are presented in Table 3. The model fit was poor, so we tried conducting EFA, which hypothesized no factor structure with the Promax rotation method, using a robust maximum likelihood estimation. Table 4 shows the results of the EFA that yielded a two-factor structure. Among HCWs and non-HCWs, Section 2 (peers) and Section 3 (team as a whole) were combined into a single factor.

Table 5 shows correlations between the scores of the PS scales and the scores of the PS scale for workers developed by Liang et al., social support at work, servant leadership, organization-based self-esteem, and organizational justice. The PS score of the full scale and all the three subscales was significantly and positively correlated with the scores of all the scales. For non-HCWs, full scale had a high correlation with PS scale for workers developed by Liang et al. (r = 0.735), with supervisor support (r = 0.729), with empowerment (r = 0.757), and with interactional justice (r = 0.723). Section 1 (team leader) had a high correlation with PS scale for workers developed by Liang et al. (r = 0.711), supervisor support (r = 0.761), empowerment (r = 0.753), standing back (r = 0.709), and interactional justice (r = 0.748). Section 3 (team as a whole) showed high correlation with empowerment (r = 0.701). HCW did not achieve high correlations (r < 0.70) but showed a similar trend to non-HCW.

The results of the MLR analyses are shown in Table 6. In HCWs, the full scale showed significant associations with low psychological distress (adjusted β = −0.508, *p* < 0.001), high work engagement (adjusted β = 0.462, *p* < 0.001), high job performance (adjusted β = 0.476, *p* < 0.001), and high job satisfaction (adjusted β = 0.592, *p* < 0.001). In Model 1 (individually entered), all three subscales of the scale (team leader, peer, and team as a whole) were significantly associated with low psychological distress, high work engagement, high job performance, and high job satisfaction. In Model 2 (simultaneously entered), Section 1 (team leader) was significantly associated with high work engagement, high job performance, and high job satisfaction in the adjusted model. Section 2 (peers) was significantly associated with low psychological distress. Section 3 (team as a whole) was significantly associated with high job satisfaction.

For non-HCWs, the full scale showed significant associations with low psychological distress (adjusted β = −0.424, *p* < 0.001), high work engagement (adjusted β = 0.510, *p* < 0.001), high job performance (adjusted β = 0.494, *p* < 0.001), and high job satisfaction (adjusted β = 0.587, *p* < 0.001). In Model 1 (individually entered), all three subscales showed significant associations similar to those observed in HCWs. In Model 2 (simultaneously entered), Section 1 was significantly associated with high work engagement, high job performance, and high job satisfaction in the adjusted model. Section 3 (team as a whole) was associated with high work engagement and job satisfaction. No section showed a significant association with low psychological distress in the adjusted model, but Section 1 in the crude model did show significance.

## 4. Discussion

The Japanese version of the survey measure of PS developed by O’Donovan et al. demonstrated acceptable high internal consistency, test–retest reliability, and convergent validity. Structural validity remained an issue. The full survey measure of PS showed significant associations with low psychological distress, high work engagement, high job performance, and high job satisfaction. These results were found for both HCWs and non-HCWs. Overall, the Japanese version of the survey measure of PS proved to be reliable and valid for use in all working populations.

In terms of internal consistency, Cronbach’s alpha of the full scale exceeded the stringent criterion of 0.80 [35]. The ICC for test–retest (two weeks) reliability was acceptable, except for HCWs in Section 3 (team as a whole). Because Section 3 had a small number of items, discrepancies in the evaluation of one item may easily be reflected in a lower ICC.

In CFA, the three-factor model did not have a good fit theoretically. The indicators of the fit model in CFA showed a low to moderately acceptable fit of the three-factor model. Rather, EFA suggested a two-factor structure. Peers and team as a whole were combined into one factor, suggesting that the Japanese population might imagine colleagues (peers) when they see the word “team”. A future study is needed to examine the structure in another sample.

The factor loading pattern was almost identical for factor 1 (peers and team) among both HCWs and non-HCWs. However, the pattern differed slightly for factor 2 (leader), while “speaking up is valued by team leader” (no. 7) loaded highly on both. For HCWs, a “sense of trust in team leader” (no. 9) and “support for the new task and learning (no.8) had high loadings, while for non-HCWs, “feeling safe discussing personal problems and disagreements” (no. 3) and “communicating about work issues” (no. 2) had high loadings. In clinical settings, patient safety and speaking are likely to be prioritized regardless of leaders’ attitudes. While leaders’ behavioral integrity affected the reported treatment errors [36], trust in leaders may influence the PS atmosphere among Japanese HCWs. Support for learning new tasks may characterize leaders who create psychologically safe workplaces in Japanese clinical settings. In non-HCWs, a previous study suggested that being allowed to express opinions and doing so were different experiences among Japanese workers [12]. Leaders’ willingness to allow and encourage employees to speak up and employees’ perceptions of doing so may both be required to ensure PS among non-HCWs.

Convergent validities were also well supported, as we expected. The findings were in line with previous research showing the positive association of PS with supervisor support, co-workers’ support, and organizational factors [12]. A supportive work environment may make workers feel safe in taking interpersonal risks. PS has been known to mediate the relationship between servant or inclusive leadership and job-related outcomes (e.g., job performance) [5,6,7,8]. Concerning servant leadership, subscales of empowerment showed the greatest associations for both HCWs and non-HCWs. Empowerment in leadership was defined as a motivational concept aimed at fostering a pro-active, self-confident attitude among followers and giving them a sense of personal power by encouraging self-directed decision making, information sharing, and coaching for innovative performance [24]. In Japan, leaders who can empower their team members also facilitate PS. For non-HCWs, PS was highly correlated (r > 0.70) with supervisor factors, such as supervisor support, leadership (especially empowerment), and interactional justice. For HCWs, no measure achieved high correlations. The leader’s supportive attitude, examined in previous research, may correspond with PS for non-HCW, and other workplace factors may influence clinical settings. Another reason may be that measurement scales tested for convergent validity were developed for workers (not specifically for HCWs). Overall, theoretical associations suggested good convergent validity for both HCWs and non-HCWs.

The full scale of the survey measure of PS was significantly associated with low psychological distress, high work engagement, high job performance, and high job satisfaction, as we expected. This finding empirically demonstrated the theoretical framework stated in the previous literature [1]. Model 2 (simultaneous entry) showed significant associations between Section 1 (team leader) and work engagement, job performance, and job satisfaction for both HCWs and non-HCWs. Given the Japanese corporate culture that emphasizes hierarchical relationships [37], the team leader may be listening to and respecting others to enhance these job-related positive outcomes. At the same time, low psychological distress was significantly associated with Section 2 (peers) only for HCWs. As mentioned earlier, speaking up is especially important in clinical settings to prioritize patient safety [36]; therefore, for HCWs, an environment where they cannot admit their mistakes or point out those of their peers may cause frustration and psychological distress. A previous study reported that the ability of nurses to forgive themselves and others was significantly associated with PS [38]. Lack of PS from peers may increase the risk of mental health deterioration among HCWs. Peers’ role may be more essential for mental health in clinical settings than in other workplaces. PS was associated with high work engagement and job performance in this study. A safe atmosphere where workers can ask questions, communicate opinions, raise issues, and suggest new ideas may increase their motivation.

This study had several limitations. It was conducted online, and participants were recruited from the research company panel, decreasing the generalizability. In addition, the self-reporting style could have biased the results; for example, people with high distress may have rated the items differently. Finally, the cross-sectional nature of the analysis precluded the assessment of causal relationships. Future studies could explore the associations of PS with outcomes using longitudinal design and workers from more diverse backgrounds.

## 5. Conclusions

The Japanese version of the survey measure of PS developed by O’Donovan et al. had acceptable reliability and validity for both HCWs and non-HCWs groups, while structural validity remained an issue and needs further examination. This measure is the first Japanese scale that can evaluate the multidimensional PS of leaders, peers, and teams in the workplace. The associations with other important factors [2] (e.g., creativity, learning behavior) and the mediator role of PS, which recent studies examined [5,6,7,8,9,10,11], were not investigated in this study. Such evidence should be replicated in the future, using this scale in Japan. Despite the limitation of the cross-sectional analysis, PS showed positive associations with good mental health and positive job-related outcomes in this study. Considering the present findings that there was a slight difference in impacts of PS in HCWs and non-HCWs on employees’ mental health, future research may be able to develop effective interventions to improve PS by industry. Examining multiple aspects of PS may also improve the workplace environment by considering specific issues in each workplace context.

## Figures and Tables

**Table 1 ijerph-19-09879-t001:** Characteristics of Japanese non-manager employees with more than three team members.

	Healthcare Workers (HCW)		Non-HCW	
	Baseline (n = 200)	Follow-Up (n = 100)	Baseline (n = 200)	Follow-Up (n = 100)
	n (%)/Mean (SD)	n (%)/Mean (SD)	n (%)/Mean (SD)	n (%)/Mean (SD)
Gender				
Men	80 (40.0)	41 (41.0)	138 (69.0)	67 (67.0)
Women	120 (60.0)	59 (59.0)	62 (31.0)	33 (33.0)
Age (year)	40.1 (9.6)	40.8 (9.5)	43.4 (10.7)	43.9 (10.3)
Marital status				
Single	66 (33.0)	27 (27.0)	70 (35.0)	37 (37.0)
Married	116 (58.0)	65 (65.0)	114 (57.0)	54 (54.0)
Divorced/widowed	18 (9.0)	8 (8.0)	16 (8.0)	9 (9.0)
Educational attainment				
High school or less	5 (2.5)	5 (5.0)	50 (25.0)	23 (23.0)
Junior college/vocational school	78 (39.0)	42 (42.0)	26 (13.0)	15 (15.0)
University or higher	117 (58.5)	53 (53.0)	124 (62.0)	62 (62.0)
Occupation				
Professional/technician	180 (90.0)	94 (94.0)	54 (27.0)	32 (32.0)
Clerical	8 (4.0)	4 (4.0)	74 (37.0)	37 (37.0)
Manual workers	4 (2.0)	1 (1.0)	25 (12.5)	10 (10.0)
Service workers	1 (0.5)	0 (0.0)	42 (21.0)	19 (19.0)
Others	7 (3.5)	1 (1.0)	5 (2.5)	2 (2.0)
Type of healthcare worker				
Physicians	28 (14.0)	12 (12.0)	n/a	n/a
Nurses	95 (47.5)	47 (47.0)	n/a	n/a
Others	77 (38.5)	41 (41.0)	n/a	n/a
Company size				
1000 or more	73 (36.5)	31 (31.0)	82 (41.0)	39 (39.0)
500–999	25 (12.5)	13 (13.0)	16 (8.0)	10 (10.0)
300–499	35 (17.5)	21 (21.0)	18 (9.0)	10 (10.0)
100–299	38 (19.0)	19 (19.0)	31 (15.5)	14 (14.0)
50–99	8 (4.0)	1 (1.0)	23 (11.5)	13 (13.0)
20–49	4 (2.0)	2 (2.0)	15 (7.5)	7 (7.0)
5–19	17 (8.5)	13 (13.0)	15 (7.5)	7 (7.0)
Number of team members				
20 or more	89 (44.5)	40 (40.0)	26 (13.0)	12 (12.0)
11–19	46 (23.0)	24 (24.0)	30 (15.0)	12 (12.0)
6–10	41 (20.5)	21 (21.0)	87 (43.5)	46 (46.0)
3–5	24 (12.0)	15 (15.0)	57 (28.5)	30 (30.0)
Status of team leader				
Manager	79 (39.5)	36 (36.0)	89 (44.5)	46 (46.0)
Not a manager	121 (60.5)	64 (64.0)	111 (55.5)	54 (54.0)
Working style				
Commuting	198 (99.0)	98 (98.0)	134 (67.0)	64 (64.0)
Working from home (WFH)	0 (0.0)	0 (0.0)	15 (7.5)	9 (9.0)
Hybrid	1 (0.5)	1 (1.0)	50 (25.0)	27 (27.0)
Other	1 (0.5)	1 (1.0)	1 (0.5)	0 (0.0)

SD: standard deviation.

**Table 2 ijerph-19-09879-t002:** The mean scores of the survey measures of psychological safety and internal and test–retest reliability.

	HCW				Non-HCW			
	Baseline (n= 200)		Follow-Up (n = 100)		Baseline (n = 200)		Follow-Up (n = 100)	
Subscales [Possible Range]	Mean (SD)	Cronbach’s α	Mean (SD)	ICC	Mean (SD)	Cronbach’s α	Mean (SD)	ICC
Section 1 (team leader) [1–7]	4.89 (1.32)	0.95	4.76 (1.24)	0.89	4.76 (1.39)	0.96	4.58 (1.50)	0.92
Section 2 (peers) [1–7]	5.04 (1.26)	0.94	4.90 (1.20)	0.83	4.71 (1.41)	0.96	4.73 (1.51)	0.84
Section 3 (team as a whole) [1–7]	4.98 (1.36)	0.91	4.80 (1.24)	0.75	4.59 (1.50)	0.93	4.58 (1.59)	0.90
Full scale [1–7]	4.96 (1.17)	0.96	4.82 (1.11)	0.88	4.71 (1.28)	0.97	4.63 (1.40)	0.92

HCW: healthcare workers. ICC: intra-class correlation coefficient.SD: standard deviation.

**Table 3 ijerph-19-09879-t003:** Factor loading scores from the confirmatory factor analysis based on three-factor model.

	Factor Loading Scores	
	HCW (Baseline n = 200)	Non-HCW (Baseline n = 200)
Section 1 (team leader)		
1 If I had a question or was unsure of something in relation to my role at work, I could ask my team leader.	0.81	0.80
2 I can communicate my opinions about work issues with my team leader.	0.88	0.85
3 I can speak up about personal problems or disagreements to my team leader.	0.78	0.85
4 I can speak up with recommendations/ideas for new projects or changes in procedures to my team leader.	0.84	0.86
5 If I made a mistake on this team, I would feel safe speaking up to my team leader.	0.83	0.87
6 If I saw a colleague making a mistake, I would feel safe speaking up to my team leader	0.81	0.82
7 If I speak up/voice my opinion, I know that my input is valued by my team leader.	0.87	0.92
8 My team leader encourages and supports me to take on new tasks or to learn how to do things I have never done before.	0.86	0.85
9 If I had a problem in this company, I could depend on my team leader to be my advocate.	0.89	0.84
Section 2 (peers)		
1 If I had a question or was unsure of something in relation to my role at work, I could ask my peers.	0.82	0.79
2 I can communicate my opinions about work issues with my peers.	0.86	0.88
3 I can speak up about personal issues to my peers.	0.73	0.76
4 I can speak up with recommendations/ideas for new projects or changes in procedures to my peers.	0.89	0.90
5 If I made a mistake on this team, I would feel safe speaking up to my peers.	0.88	0.94
6 If I saw a colleague making a mistake, I would feel safe speaking up to this colleague.	0.85	0.90
7 If I speak up/voice my opinion, I know that my input is valued by my peers.	0.86	0.92
Section 3 (team as a whole)		
1 It is easy to ask other members of this team for help.	0.87	0.95
2 People keep each other informed about work-related issues in the team.	0.95	0.90
3 There are real attempts to share information throughout the team.	0.83	0.86

HCW: healthcare workers.

**Table 4 ijerph-19-09879-t004:** Exploratory factor analysis without assuming the number of factors by using maximum likelihood method with Promax rotation.

	Factor Loading Score	
	Factor 1	Factor 2
HCW (baseline n = 200)		
(peers) 5 If I made a mistake on this team, I would feel safe speaking up to my peers.	**0.927**	−0.061
(peers) 2 I can communicate my opinions about work issues with my peers.	**0.921**	−0.096
(peers) 4 I can speak up with recommendations/ideas for new projects or changes in procedures to my peers.	**0.846**	0.043
(peers) 1 If I had a question or was unsure of something in relation to my role at work, I could ask my peers.	**0.813**	0.012
(peers) 3 I can speak up about personal issues to my peers.	**0.812**	−0.105
(peers) 6 If I saw a colleague making a mistake, I would feel safe speaking up to this colleague.	**0.794**	0.069
(peers) 7 If I speak up/voice my opinion, I know that my input is valued by my peers	**0.779**	0.106
(team as a whole) 2 People keep each other informed about work-related issues in the team.	**0.725**	0.167
(team as a whole) 1 It is easy to ask other members of this team for help.	**0.645**	0.180
(team as a whole) 3 There are real attempts to share information throughout the team.	**0.519**	0.295
(team leader) 9 If I had a problem in this company, I could depend on my team leader to be my advocate.	−0.064	**0.948**
(team leader) 7 If I speak up/voice my opinion, I know that my input is valued by my team leader.	−0.092	**0.946**
(team leader) 8 My team leader encourages and supports me to take on new tasks or to learn how to do things I have never done before.	0.030	**0.848**
(team leader) 6 If I saw a colleague making a mistake, I would feel safe speaking up to my team leader.	−0.029	**0.832**
(team leader) 4 I can speak up with recommendations/ideas for new projects or changes in procedures to my team leader.	0.065	**0.778**
(team leader) 1 If I had a question or was unsure of something in relation to my role at work, I could ask my team leader.	0.036	**0.778**
(team leader) 2 I can communicate my opinions about work issues with my team leader.	0.071	**0.747**
(team leader) 5 If I made a mistake on this team, I would feel safe speaking up to my team leader.	0.141	**0.728**
(team leader) 3 I can speak up about personal problems or disagreements to my team leader	0.093	**0.703**
Non-HCW (baseline n = 200)		
(peers) 6 If I saw a colleague making a mistake, I would feel safe speaking up to this colleague.	**0.975**	−0.109
(peers) 5 If I made a mistake on this team, I would feel safe speaking up to my peers.	**0.960**	−0.037
(peers) 4 I can speak up with recommendations/ideas for new projects or changes in procedures to my peers.	**0.886**	0.018
(peers) 7 If I speak up/voice my opinion, I know that my input is valued by my peers.	**0.880**	0.048
(peers) 3 I can speak up about personal issues to my peers.	**0.863**	−0.144
(peers) 2 I can communicate my opinions about work issues with my peers.	**0.844**	0.033
(peers) 1 If I had a question or was unsure of something in relation to my role at work, I could ask my peers.	**0.777**	0.013
(team as a whole) 1 It is easy to ask other members of this team for help.	**0.679**	0.271
(team as a whole) 2 People keep each other informed about work-related issues in the team.	**0.661**	0.239
(team as a whole) 3 There are real attempts to share information throughout the team.	**0.611**	0.221
(team leader) 3 I can speak up about personal problems or disagreements to my team leader.	−0.131	**0.952**
(team leader) 7 If I speak up/voice my opinion, I know that my input is valued by my team leader.	−0.008	**0.929**
(team leader) 2 I can communicate my opinions about work issues with my team leader.	−0.022	**0.881**
(team leader) 1 If I had a question or was unsure of something in relation to my role at work, I could ask my team leader.	−0.098	**0.875**
(team leader) 4 I can bring recommendations/ideas for new projects or changes in procedures to my team leader.	0.013	**0.856**
(team leader) 5 If I made a mistake on this team, I would feel safe speaking up to my team leader.	0.061	**0.829**
(team leader) 8 My team leader encourages and supports me to take on new tasks or to learn how to do things I have never done before.	0.128	**0.750**
(team leader) 6 If I saw a colleague making a mistake, I would feel safe speaking up to my team leader.	0.145	**0.708**
(team leader) 9 If I had a problem in this company, I could depend on my team leader to be my advocate.	0.184	**0.696**

Note: Bold-faced font emphasized the larger loading scores between Factor 1 and 2.

**Table 5 ijerph-19-09879-t005:** Pearson’s correlation coefficients between each subscale on the psychological safety scale and other psychometric scales (convergent validity).

	HCW (n = 200)				Non-HCW (n = 200)			
Scales [Possible Range]	Full Scale	Section 1 (Team Leader)	Section 2 (Peers)	Section 3 (Team as a Whole)	Full Scale	Section 1 (Team Leader)	Section 2 (Peers)	Section 3 (Team as a Whole)
Psychological Safety Scale for Workers [1–5]	0.657 *	0.628 *	0.536 *	0.603 *	0.735 *	0.711 *	0.589 *	0.700 *
Social support at work (BJSQ)								
Supervisor support [1–4]	0.640 *	0.696*	0.425 *	0.553 *	0.729 *	0.761 *	0.537 *	0.647 *
Coworkers support [1–4]	0.557 *	0.389 *	0.612 *	0.593 *	0.672 *	0.501*	0.694 *	0.715 *
Servant leadership survey								
Empowerment [1–6]	0.655 *	0.680 *	0.481 *	0.560 *	0.757 *	0.753 *	0.589 *	0.701 *
Humility [1–6]	0.494 *	0.547 *	0.315 *	0.428*	0.644 *	0.654 *	0.500*	0.567 *
Standing back [1–6]	0.564 *	0.609 *	0.384 *	0.486 *	0.694 *	0.709 *	0.538 *	0.597 *
Stewardship [1–6]	0.574 *	0.580 *	0.440 *	0.496 *	0.625 *	0.595 *	0.525 *	0.573*
Authenticity [1–6]	0.572 *	0.616 *	0.398 *	0.471 *	0.660 *	0.649 *	0.538 *	0.581 *
Organization-based self-esteem [1–5]	0.421 *	0.387 *	0.403 *	0.306 *	0.529 *	0.477 *	0.466 *	0.512 *
Organizational justice								
Procedural justice [1–5]	0.570 *	0.586 *	0.419 *	0.505 *	0.594 *	0.586 *	0.471 *	0.548 *
Interactional justice [1–5]	0.596 *	0.654 *	0.397 *	0.501 *	0.723 *	0.748 *	0.547 *	0.629 *

HCW: healthcare workers; BJSQ: Brief Job Stress Questionnaire; * *p* < 0.01.

**Table 6 ijerph-19-09879-t006:** Associations of psychological safety scale with psychological distress, work engagement, job performance, and job satisfaction.

	Psychological Distress (K6)				Work Engagement (UWES-9)				Job Performance (HPQ)				Job Satisfaction (BJSQ)			
	Crude		Adjusted ^(c)^		Crude		Adjusted ^(c)^		Crude		Adjusted ^(c)^		Crude		Adjusted ^(c)^	
Variables	β	*p*	β	*p*	β	*p*	β	*p*	β	*p*	β	*p*	β	*p*	β	*p*
HCWs																
Full scale	−0.507	<0.001 *	−0.508	<0.001 *	0.465	<0.001 *	0.462	<0.001 *	0.476	<0.001 *	0.476	<0.001 *	0.597	<0.001 *	0.592	<0.001 *
Model 1 ^(a)^																
Section 1 (team leader)	−0.422	<0.001 *	−0.431	<0.001 *	0.428	<0.001 *	0.422	<0.001 *	0.479	<0.001 *	0.477	<0.001 *	0.542	<0.001 *	0.543	<0.001 *
Section 2 (peers)	−0.508	<0.001 *	−0.497	<0.001 *	0.409	<0.001 *	0.413	<0.001 *	0.390	<0.001 *	0.390	<0.001 *	0.500	<0.001 *	0.495	<0.001 *
Section 3 (team as a whole)	−0.448	<0.001 *	−0.445	<0.001 *	0.411	<0.001 *	0.409	<0.001 *	0.366	<0.001 *	0.381	<0.001 *	0.605	<0.001 *	0.590	<0.001 *
Model 2 ^(b)^																
Section 1 (team leader)	−0.128	0.141	−0.138	0.131	0.243	0.008 *	0.210	0.026 *	0.396	<0.001 *	0.365	<0.001 *	0.251	0.002 *	0.245	0.003 *
Section 2 (peers)	−0.363	0.001 *	−0.332	0.002 *	0.140	0.193	0.162	0.140	0.131	0.219	0.106	0.309	−0.030	0.754	−0.013	0.889
Section 3 (team as a whole)	−0.075	0.473	−0.093	0.384	0.135	0.215	0.143	0.195	−0.006	0.955	0.056	0.590	0.459	<0.001*	0.439	<0.001
Non-HCWs																
Full scale	−0.458	<0.001 *	−0.424	<0.001 *	0.524	<0.001 *	0.510	<0.001 *	0.516	<0.001 *	0.494	<0.001 *	0.598	<0.001 *	0.587	<0.001 *
Model 1 ^(a)^																
Section 1 (team leader)	−0.405	<0.001 *	−0.372	<0.001 *	0.504	<0.001 *	0.496	<0.001 *	0.498	<0.001 *	0.484	<0.001 *	0.580	<0.001 *	0.574	<0.001 *
Section 2 (peers)	−0.422	<0.001 *	−0.391	<0.001 *	0.413	<0.001*	0.395	<0.001*	0.425	<0.001 *	0.397	<0.001 *	0.479	<0.001 *	0.467	<0.001 *
Section 3 (team as a whole)	−0.422	<0.001 *	−0.396	<0.001 *	0.522	<0.001 *	0.509	<0.001 *	0.474	<0.001 *	0.454	<0.001 *	0.567	<0.001 *	0.552	<0.001 *
Model 2 ^(b)^																
Section 1 (team leader)	−0.185	0.049 *	−0.152	0.103	0.280	0.002 *	0.278	0.002 *	0.318	<0.001 *	0.318	<0.001 *	0.361	<0.001 *	0.362	<0.001 *
Section 2 (peers)	−0.195	0.086	−0.172	0.133	−0.104	0.327	−0.137	0.209	0.045	0.677	0.006	0.959	−0.032	0.750	−0.035	0.731
Section 3 (team as a whole)	−0.127	0.304	−0.146	0.241	0.405	<0.001 *	0.423	<0.001 *	0.207	0.082	0.222	0.064	0.332	0.003 *	0.322	0.004 *

^(a)^ Three subscales of psychological safety scale (team leader, peer, and team as a whole) were individually entered. ^(b)^ Three subscales of psychological safety scale (team leader, peer, and team as a whole) were simultaneously entered. ^(c)^ The adjusted model additionally adjusted for sex, age, industry, type of healthcare worker, working style (e.g., work from home), educational attainment, company size, and occupation among HCWs, and adjusted for the same variables excluding type of healthcare workers among non-HCWs. K6: Kessler 6; UWES: Utrecht Work Engagement Scale; HCW: healthcare worker; HPQ: Health and Work Performance Questionnaire; BJSQ: Brief Job Stress Questionnaire; * *p* < 0.05.

## Data Availability

The data supporting this study’s findings are available from the corresponding author, KI, upon reasonable request.

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
