# Peer review of "The Survey Measure of Psychological Safety and Its Association with Mental Health and Job Performance: A Validation Study and Cross-Sectional Analysis"

_ijerph, 2022, doi:10.3390/ijerph19169879_

Round 1
Reviewer 1 Report
I have reviewed your article and my suggestions are presented below:
The introduction lacks the rationale of the study. Authors must provide the reasons as to why this research is needed and what issues it resolves. The objective of the study should be explained in more detail. The authors have mentioned only two objectives, however, the research encompasses more than that. Therefore, the authors should identify and mention all the objectives. 'scale information' should be presented in the relevant section, not in the introduction.
Need theoretical foundation and reason why the variables are being combined. Theory structures your research and it explains why these constructs are being investigated and measured. The authors should explain the constructs with regard to some theories.
In the Method section, the authors have mentioned that they used 'multiple linear regression analyses to examine the relationship...' but did not identify the techniques that were used to guide the analysis. What procedures were followed, how were they followed, and what criteria were used to make decisions. All this must be supported by the literature with sources provided. This part of the Method section is missing. The authors should also explain why have they choose multiple linear regression analysis to test the relationship.
The authors have explained the results in the Discussion section. They should write about the results in the Result section. The Discussion section should be focused on comparing your findings with other studies and in-depth analysis of any unexpected or profound results. The implications of the research are also missing. The authors must write about the implications of their study. What real-world problem is solved and how this study could be beneficial for the practitioners.
Reviewer 2 Report
1. The paper does not meet the journal standard for referencing sources in the text.
2. There should be a short theoretical presentation of the multiple linear regression model used in the application.
3. Research methodology is not very clearly defined, it needs further explanation of the need for using statistical methods.
4. I suggest to the authors to present and coment the statistical results in the context of their application.
5. I suggest to the authors to further develop their literature review section, including current sources, from 2022, from the web of Science Category.
Round 2
Reviewer 2 Report
I have read the submitted material and have a few comments I would like to address to the authors:
1. Tables 1-6 do not exist in the article, they are referred in the text, but they are not. I suggest the authors to include them in the paper.
2. In the supplementary files section there is Apendix 1 in a different language than English. I suggest to the authors to translate that material fully into English.
3. The abstract should be about 200 words maximum. Authors should reduce the number of words of it.
4. The article has too many subheadings, it is too much fragmented. I suggest the authors to structure the research paper by research sections that they find in the instructions for authors.
5. Line 307, model 1 is used, but in the text is not referred to anywhere.
6. I suggest to the authors that the limitations section of the study be included in the conclusion.
7. In the conclusion I consider to be necessary that the authors to include the future research section, the novelty of the study and the gap from the literature that the study covers.
Round 3
Reviewer 2 Report
I have read the submitted material and have a few comments that I would like to address to the authors:
1. In the supplementary files section there is Apendix 1 in a language other than English. I suggest to the authors to translate the whole material into English, since the journal is an international one, possibly also let the both languages: english and the other.
2. Lines 102,181, 204,209,221, I do not understand their purpose.
3. What does PS mean? It is used in the text, but is not explained anywhere.
4. Section 2 is much too fragmented. I suggest that 2.1, 2.2, 2.3 be in one section. I also suggest the authors to elaborate a bit more on the "statistical analysis" part.
